# Bioprospecting for Bioactive Peptide Production by Lactic Acid Bacteria Isolated from Fermented Dairy Food

**Davide Tagliazucchi** , **Serena Martini and Lisa Solieri ***

Department of Life Sciences, University of Modena and Reggio Emilia, Via Amendola 2, 42122 Reggio Emilia, Italy; davide.tagliazucchi@unimore.it (D.T.); serena.martini@unimore.it (S.M.)
* Correspondence: lisa.solieri@unimore.it; Tel.: +39-0522-2026

**Abstract:** With rapidly ageing populations, the world is experiencing unsustainable healthcare from chronic diseases such as metabolic, cardiovascular, neurodegenerative, and cancer disorders. Healthy diet and lifestyle might contribute to prevent these diseases and potentially enhance health outcomes in patients during and after therapy. Fermented dairy foods (FDFs) found their origin concurrently with human civilization for increasing milk shelf-life and enhancing sensorial attributes. Although the probiotic concept has been developed more recently, FDFs, such as milks and yoghurt, have been unconsciously associated with health-promoting effects since ancient times. These health benefits rely not only on the occurrence of fermentation-associated live microbes (mainly lactic acid bacteria; LAB), but also on the pro-health molecules (PHMs) mostly derived from microbial conversion of food compounds. Therefore, there is a renaissance of interest toward traditional fermented food as a reservoir of novel microbes producing PHMs, and "hyperfoods" can be tailored to deliver these healthy molecules to humans. In FDFs, the main PHMs are bioactive peptides (BPs) released from milk proteins by microbial proteolysis. BPs display a pattern of biofunctions such as anti-hypertensive, antioxidant, immuno-modulatory, and anti-microbial activities. Here, we summarized the BPs most frequently encountered in dairy food and their biological activities; we reviewed the main studies exploring the potential of dairy microbiota to release BPs; and delineated the main effectors of the proteolytic LAB systems responsible for BPs release.

**Keywords:** fermented dairy food; bioactive peptides; *Lactobacillus*; *Lactococcus*; *Streptococcus*; antihypertensive peptides; antioxidant peptides; anti-microbial peptides; anti-diabetic peptides; proteolytic system

---

## 1. Introduction

The promotion of healthy and sustainable diets is one of major actions prioritized in European research agenda for supporting public health and citizens' wellbeing. Several evidences proved that some foods or food ingredients, including living microbial cells contained therein, can enhance short-term wellbeing and reduce initiation and/or progression of non-diseases and the associated chronic conditions (e.g., musculoskeletal disorders, cardiovascular diseases and stroke, hypertension, obesity, type II diabetes mellitus (T2D), cancers, or mental health conditions) [1,2]. In view of the importance of nutrition in health promotion, academia and the food industry face the new challenge to tailor novel strategies and products that better contribute to human welfare as well as reducing the risk of insurgence of specific pathologies [3]. The term "functional food," was coined in Japan and the USA in the 1970s, to indicate food products fortified with special constituents that possess advantageous physiological effects supported in vitro and in vivo [4]. Generally, functional food is

deprived of some "dangerous" components, such as salt, sugars, and saturated fat levels, and/or is enriched in probiotic cells and/or pro-health molecules (PHMs). Although functional foods are not defined legally, scientific literature regarding these products is increasing to demonstrate that they provide additional benefits beyond the nutrient intake and the hunger satisfaction [5,6]. However, much more effort should be done yet to identify PHMs and to elucidate their multiple mechanisms of action in the food–human consortium. This knowledge is essential to design a novel generation of functional food with the maximally positive health impact, the so-called "hyperfood" (Figure 1) [7].

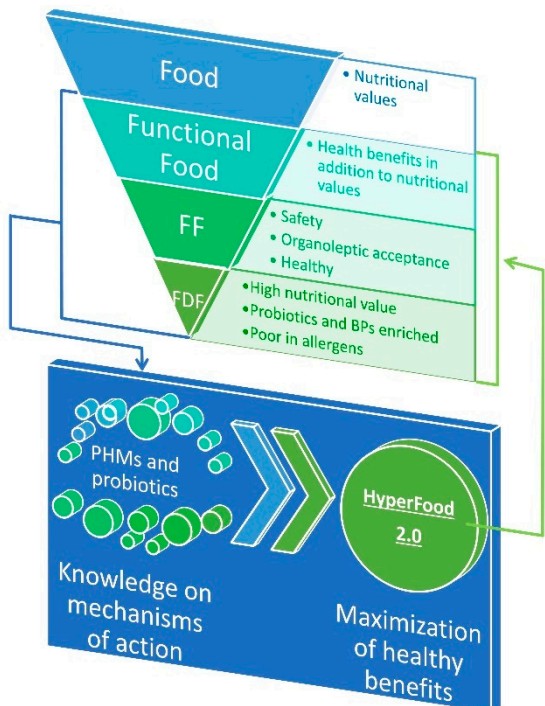

**Figure 1.** Position of fermented dairy food (FDF) in food consortium and strategies for knowledge-driven optimization of a novel generation of functional food, called HyperFood 2.0. Abbreviations: FF, fermented food; FDF, fermented dairy food; BPs, bioactive peptides; PHMs, pro-health molecules.

Food fermentation is generally defined as the microbial conversion of food macro- and micro-compounds under anaerobic conditions. Since the early phases of civilization, mankind unconsciously utilized microbes for preserving food and enhancing flavors, aromas, and textures [7,8]. Repetition of these practices over thousands of years lead to the adaptation of microorganisms to food ecosystems [9]. Today, one-third of the human diet worldwide still consists of fermented foods and beverages (generally referred as FF) [10]. In industrializing countries, well-established fermentation processes, involving selected starter cultures or microbial consortia, assure safety and organoleptic properties highly appreciated by consumers. However, FF produced by indigenous microbiota or back-slopping procedure still remains relevant in artisanal segments of the market both in Western and in Eastern countries. Association between FF and health dated back from ancient Greek, when Hippocrates formulated the concept of "food as medicine". Fermented dairy food (FDF) is a generic term used to define three main types of dairy food matrices obtained through lactic fermentation by lactic acid bacteria (LAB): Liquid (fermented milks), semi-solid (yogurt and some fresh cheeses), and solid (the majority of cheeses). FDFs have been the paradigm of healthy food since Metchnikoff's studies [11–14] (Figure 1). Recently, randomized controlled trials, prospective cohort studies, and meta-analyses supported an inverse relation between consumption of FDFs and overweight [15–18] or risk of metabolic syndrome [19,20], cardiovascular disease [21–23], T2D [24–26], and overall mortality [27]. These data should be considered with caution for at least three reasons: the variety of dairy products considered can be a confounding element, which biases inter-studies

comparison; dosage has not been harmonized across different experimental cohorts; and the impact of other factors, such as gender and age, has not been considered. Nevertheless, these evidences encourage the uptake of FDF and promote the search for molecular effectors underpinning FDFs-associated healthy benefits, which, in turn, can drive the translation towards more reliable health benefits for improved FDF.

In case of FDF, milk itself provides a large number of beneficial compounds. According to the consensus document published in May 2017 by a panel of 18 experts, these PHMs include proteins, minerals (calcium, phosphorus, and magnesium), water-soluble vitamins (B12 and riboflavin), fat-soluble vitamins (A, D, and K2), and specific bioactive lipids [28]. In addition to these nutritional attributes, fermentation processes provide other promoting healthy effects. FDF delivers specifically selected or indigenous probiotic microbes into the human intestine, where they may modulate immune response, enhance epithelial barrier function, or modify indigenous gut microbiota [29,30]. Furthermore, during lactic fermentation certain potentially "dangerous" compounds are depleted by microbial conversion, e.g., degradation of allergenic α-lactalbumin and β-lactoglobulin into oligopeptides or fermentation of lactose into lactic acid [31]. Finally, microbes can release secondary metabolites, which act as PHMs in human body [32].

Several reviews revised different pro-health aspects of milk and FDF as well as their role as carrier for probiotics [33–38]. Here, we focused on bioactive peptides (BPs), the most relevant PHMs released from milk proteins by the action of LAB inhabiting FDF. We summarized the most compelling evidences on how FDFs are bio-reservoirs of novel BPs-producing LAB and highlighted how their usage is relevant in developing novel functional food with maximized amount of BPs through an approach referred to as reverse food engineering.

## 2. Survey of BPs in FDF

BPs are peculiar protein fragments that modulate various regulatory processes at different levels in the human body, such as digestive, immune, cardiovascular, endocrine, and nervous systems, ultimately influencing human health [39,40]. To date, milk caseins are viewed as the most important source of BPs and products containing milk-derived peptides have an increasing market trend (Table 1). Although exogenous enzymes or proteolysis during gastro-intestinal (GI) transit can also release BPs, we only focused on BPs produced by LAB proteolysis.

Numerous BPs are detected in FF and are reported as important agents for different functional activities such as antioxidative, antihypertensive, antimicrobial, immunomodulatory, opioid, hypocholesterolemic, mineral binding, and bone mineralization activities.

To attend the above-mentioned effects in vivo, these BPs should reach intact the target site, and thus remain active after oral ingestion and during GI digestion and absorption [41,42]. This implies resistance to enzymatic hydrolysis by gastric and intestinal proteases and peptidases laying along the brush border membrane of the villi. When absorbed, short peptides (usually di- and tripeptides) are transported across intestinal enterocytes via a specific peptide transport system (PepT1), whereas largest peptides are likely absorbed by transcellular or paracellular mechanisms or via endocytosis [41,43]. Nonetheless, there are substantial evidences indicating the absorption of bioactive peptides in humans and animals, in some cases reaching blood concentrations of few μmolar [41–44]. Since the bioavailability of BPs is a relevant but often poorly explored point, in this survey we detailed whether there are in vivo evidences of healthy action for every potential BPs.

**Table 1.** Principal milk-derived products containing bioactive peptides (BPs) currently available.

| Brand | Category | Producers | Molecules | Bioactivities |
|---|---|---|---|---|
| BioPURE-GMP | Whey protein isolate | Davisco, USA | κ-casein f(106–109) | Satiety regulation, anti-carcinogenic, antimicrobial, anti-thrombotic BPs |
| BioZate | Whey protein isolate | Davisco, USA | β-lactoglobulin fragments | Hypotensive BPs |
| C12 Peption | Ingredients | DMV, The Netherlands | Casein derived FFVAPFPEVFGK | Hypotensive BP |
| Capolac | Ingredients | Arla Food, Denmark | Casein phosphopeptide | Mineral absorption |
| Casein DP/Peptio Drink | Soft milk drink | Kracie Pharmaceuticals, Japan | Casein derived FFVAPFPEVFGK | Hypotensive |
| CE90CPP | Ingredients | DMV, The Netherlands | Casein phosphopeptide | Mineral absorption |
| PeptoPro | Flavored drink | DSM Food specialties, The Netherlands | Casein phosphopeptide | Improves muscle performance |
| Cysteine peptide | Ingredients/hydrolysate | DMV, The Netherlands | Casein peptides | hypotensive BPs |
| Praventin | Capsule | DMV, The Netherlands | Lactoferrin-enriched whey protein hydrolysate | Antimicrobial BPs (reduced skin infection) |
| Festivo | Low-fat cheese | MIT Agrifood Research, Finland | αS1-casein f(1–6), αS1-casein f(1–7), αS1-casein f(1–9) | Hypotensive BPs |
| Kotsu Kotsu calcium | Soft drink | Asahi Soft Drink | Casein phosphopeptide | Mineral absorption |
| Evolus | Fermented milk | Valio Ltd. Finland | VPP, IPP | Hypotensive BPs |
| Calpis/Ameal S 120 | Sour milk | Calpis Co Ltd, Japan | VPP, IPP | Hypotensive BPs |
| Ameal S | Tablet | Calpis Co Ltd, Japan | VPP, IPP | Hypotensive BPs |
| Immunel | Ingredient | Wild Co., Germany | Milk peptides | Anti-inflammatory BPs |
| Tegricel | Ingredient | Wild Co., Germany | Milk peptides | Anti-inflammatory BPs |
| Lactium/ProDiet F200 | Flavored milk drink/capsules | Ingredia, France | αS1-casein f(91–100) YLGYLEQLLR | Stress relief BPs |
| Vivinal Alpha | Ingredient | Borculo Domo Ingredients | α-lactalbumin-rich whey protein hydrolysate | Aids relaxation and sleep |
| Dermylex | Tablet | Advitech Inc., Canada | Whey protein extract XP-828L | Reduces symptoms of psoriasis |
| Capolac | Ingredient | Arla Food Ingredients | Casein phosphopeptide | Mineral absorption |
| Recaldent | Chewing gum | Cadbury Enterprises | Calcium casein peptone-calcium phosphate | Anticariogenic BPs |

### 2.1. Anti-Hypertensive Peptides Identified in Commercial FDFs

As suggested by the European Society of Cardiology (ESC) working group on cardiovascular pharmacology and drug therapy, the nutraceutical approach for hypertension management could bring positive effects for those patients with blood pressure borderline values and may be co-administered with anti-hypertensive drugs in blood pressure reduction [45,46].

In recent years, scientists have investigated various BPs derived from FDFs, such as fermented milk, yogurt, and cheese, for their potential anti-hypertensive activity (Table 2) [40,47,48]. Different mechanisms of action were suggested to describe their anti-hypertensive effects [40,47]. The most important mechanism by which BPs may decrease blood pressure is through ACE inhibition [49,50].

**Table 2.** Peptides with in vivo anti-hypertensive activity identified in commercial FDFs.

| Peptide | Fragment | Source | In vivo Model | Dose | Systolic Blood Pressure Reduction | References |
|---|---|---|---|---|---|---|
| VPP [1] | β-casein f(84–86) | Fermented milk, cheese, yogurt | Huma SHR | 3–100 mg/day 5 mg/kg | −3.73 mmHg −20.1 mmHg | [23,51–58] |
| IPP [1] | β-casein f(74–76) κ-casein f(108–110) | Fermented milk, cheese, yogurt | Human SHR | 3–100 mg/day 5 mg/kg | −3.73 mmHg −18.3 mmHg | [23,51–58] |
| YP | Various fragments in β-casein, αS1-casein, κ-casein | Cheese, yogurt | SHR | 1 mg/kg | −27.4 mmHg | [55,59,60] |
| TKVIP | αS2-casein f(198–202) | Yogurt | SHR | 1 mg/kg | −9.2 mmHg | [55,61] |
| KVLPVPQ | β-casein A2 f(169–175) | Cheese, yogurt | SHR | 2 mg/kg | −31.5 mmHg | [55,61] |
| LHLPLP | β-casein A2 f(133–138) | Cheese, yogurt | SHR | 3 mg/kg | −25.3 mmHg | [53–55,62] |
| RYLGY | αS1-casein f(90–94) | Cheese | SHR | 5 mg/kg | −25 mmHg | [53,54,63,64] |
| AYFYPEL | αS1-casein f(143–149) | Cheese | SHR | 5 mg/kg | −20 mmHg | [53,54,63] |
| RYLG | αS1-casein f(90–93) | Cheese | SHR | 5 mg/kg | −18 mmHg | [53,54,64] |
| RY | Various fragments in αS1-casein, αS2-casein, κ-casein | Fermented milk | SHR | 5 mg/kg | −18 mmHg | [64,65] |
| HLPLP | β-casein A2 f(134–138) | Cheese | SHR | 7 mg/kg | −23.5 mmHg | [53,54,66] |
| LPLP | β-casein A2 f(135–138) | Yogurt | SHR | 7 mg/kg | −16 mmHg | [55,67] |
| PLP | β-casein A2 f(136–138) | Cheese | SHR | 7 mg/kg | −21.2 mmHg | [59,67] |
| FP | Various fragments in β-casein, αS1-casein, αS2-casein | Cheese, yogurt | SHR | 8 mg/kg | −27 mmHg | [55,58,59,68] |
| LVYPFTGPIPN | β-casein caprine f(58–68) | Kefir | SHR | 10 mg/kg | −28 mmHg | [66,69] |
| VRGPFPIIV | β-casein f(201–209) | Yogurt | SHR | 10 mg/kg | −16 mmHg | [55,62] |
| KKYNVPQL | αS1-casein caprine f(102–109) | Cheese | SHR | 10 mg/kg | −11.5 mmHg | [48,58,66] |
| AVPYPQR | β-casein f(177–183) | Kefir | SHR | 100 mg/kg | −10 mmHg | [70,71] |
| RPKHPIKHQ | αS1-casein f(1–9) | Cheese | SHR | n.a. | −9.3 mmHg | [48,58,72] |
| TPVVVPPFLQP | β-casein f(80–90) | Cheese, yogurt | SHR | n.a. | −8 mmHg | [55,59,68] |
| YPFPGPIPN | β-casein A2 f(60–68) | Cheese, kefir | SHR | n.a. | −7 mmHg | [48,58,72,73] |

[1] doses and blood pressure effect are referred to the administration of VPP + IPP in human studies.

Additional described mechanisms involve an increase in the production of the vasodilating endothelial nitric oxide (NO) as well as the inhibition of renin. Furthermore, BPs can also induce vasodilatation by reducing the activity of the sympathetic system [74].

Among the anti-hypertensive peptides found in FDFs, the lactotripeptides, VPP and IPP, gained great attention in recent years thanks to their increasingly frequent commercial use in hypotensive milk-drinks production. Indeed, numerous in vivo studies reported a blood pressure reduction in mildly hypertensive patients [23,51,75,76]. These lactotripeptides were initially detected in a sample of Japanese sour milk fermented by *Lactobacillus helveticus* and *Saccharomyces cerevisiae* and characterized for their ACE-inhibitory (ACE-i) activity and hypotensive effect in spontaneously hypertensive rats [52,77]. Next, VPP and IPP have been found in milk fermented by other LAB such as *Lactobacillus*

*casei*, *Lactobacillus delbrueckii*, and *Lactobacillus rhamnosus* as well as in commercial yogurt and several cheeses at physiological relevant concentrations [53–55,78–82]. It is worth mentioning that VPP and IPP have been also detected in physiologically relevant amounts after in vitro GI digestion of milk from different species [83–86].

In vivo effect of IPP and VPP on blood pressure in pre-hypertensive and hypertensive subjects has been recently described in numerous meta-analysis of randomized clinical trials and reviews [23,40,51,56]. Two meta-analyses of 18 and 30 placebo-controlled clinical trials found a pooled effect of lactotripeptides on systolic blood pressure reduction of −3.73 mmHg and −2.95 mmHg, respectively [23,51]. Sub-group analyses demonstrated that the decrease in systolic blood pressure in an Asian population was significantly greater (from −5.54 to −6.93 mmHg), compared with a European population (from −1.28 to −1.36 mmHg) [23,51,56,57]. Interestingly, the effect was more evident in hypertensive subjects and smaller doses (from 3 to 10 mg/day) had higher blood pressure reduction than larger dose [23,56,59,87].

Several BPs identified in commercial FDFs have demonstrated in vivo anti-hypertensive activity in spontaneously hypertensive rats (SHR) (Table 2). YP showed a potent anti-hypertensive activity in SHR decreasing the systolic blood pressure of −27.4 mmHg with a dosage of 1 mg/kg with respect to VPP and IPP, which showed a systolic blood pressure decreasing effect in SHR of −20.1 and −18.3 mmHg, respectively, at doses of 5 mg/kg [52,60]. Maeno et al. [61] isolated and identified the peptide responsible for the anti-hypertensive effect of a casein hydrolysate produced by *L. helveticus* CP790 proteinase. The purified peptide KVLPVPQ, further isolated in a commercial functional yogurt, had potent anti-hypertensive effects (−31.5 mmHg) in SHR at a dosage of 2 mg/kg [55,61]. It is important to underline that both the above described peptides displayed very low in vitro ACE-i activity, which was about 80–200 times lower than that of the lactotripeptides VPP and IPP (Table 2). These observations suggest that the assessment of the in vitro ACE-inhibitory activity of a peptide is not enough to predict or guarantee an in vivo anti-hypertensive effect [88]. This is also supported by the evidence that some BPs with very high in vitro ACE-i activity failed to exert an anti-hypertensive effect when administered to SHR [88].

The reasons for this issue could be related to (i) the resistance or susceptibility of the BPs to the hydrolysis by GI proteases, (ii) their absorption into the blood stream, and (III) different mechanisms of action other than ACE-inhibition.

Nevertheless, other peptides identified in FDFs, such as LHLPLP, RYLGY, and AYFYPEL, appeared to be potent ACE inhibitors and able to decrease systolic blood pressure in SHR even more than IPP and VPP [62,63]. The peptide LHLPLP was found in several cheeses such as Parmigiano Reggiano, Grana Padano, Cheddar, and Gorgonzola [53,54]. Interestingly, this peptide was easily released during in vitro GI digestion of Parmigiano Reggiano and Grana Padano cheeses, reaching concentration, which can show an in vivo effect [53,54]. Indeed, LHLPLP is in vitro hydrolyzed by cellular peptidases to the pentapeptide HLPLP, which still displayed high anti-hypertensive effect in SHR suggesting that this peptide is the active form of LHLPLP [66,89]. Similarly, after in vitro GI digestion, 93% of the peptide RYLGY was hydrolyzed in minor fragments, which showed anti-hypertensive activity in SHR as high as the intact parent peptide [64].

In addition, numerous ACE-i peptides have been identified in FDFs such as cheeses, yogurt, fermented milk, and kefir. Even though confirmatory studies in SHR have not yet carried out, some of these peptides exhibited low or very low IC$_{50}$ values. For example, the dipeptides IW and WL showed IC$_{50}$ values of 0.7 and 10 μmol/L, are bioavailable in humans, and were able to reduce in vivo ACE activity, appearing to be excellent candidates for further in vivo studies [90]. Furthermore, the ACE-i β-casein-derived peptides YQEPVLGPVRGPFPIIV and QEPVLGPVRGPFPIIV have been detected in plasma of human subjects after a cheese-enriched diet [44].

The complete list of ACE-i peptides together with the fermented food sources can be found in the online Supplementary Materials (Table S1).

## 2.2. Antioxidant Peptides Identified in Commercial Fermented Dairy Products

The side effects of oxidation in the body and in foodstuffs is well reported and studied. Excessive free radicals and other reactive oxygen species may affect the food quality, bringing out a range of defects such as unacceptable taste and flavor and prejudicing the shelf life [32,91]. In addition, it has been recognized a relation between age-related diseases and the presence of oxidative damages. The lipid hydroperoxides and low molecular weight compounds arising from oxidative reactions are considered the causative factors involved in these diseases. Evidences of significant free radical mediated injury to several pathological conditions are widely reported in scientific literature, including neurodegenerative disorders, atherosclerosis, diabetes, inflammatory processes, rheumatoid arthritis, and cancer [92–100]. The opportunity to reduce oxidative stress in the body through the daily intake of antioxidants, for example antioxidative peptides, may be of pivotal interest and an attractive strategy to reduce inflammatory conditions mediated by radical oxygen species (ROS) in the living cells.

Milk and dairies, as widely consumed products, are thoroughly investigated for their physiological and biochemical functions. They contain a wide range of biologically active compounds in varying proportions depending on the animal species (i.e., bovine, buffalo, goat, sheep, and camel), matrix type (i.e., milks, cheeses, fermented milks, and yogurts), and manufactural processes (i.e., mechanical, heating, and fermentative). This class of active compounds include both hydrophilic and lipophilic antioxidants such as proteins (specifically casein), small peptides, coenzyme Q10, vitamins (A, C, E, and D3), carotenoids, some minerals, and elements in trace [101,102]. Cichosz et al. [101] found that milk antioxidants synergistically worked by forming an antioxidant network, improving the antioxidant ability of milk, and protecting milk itself and its fat fraction against the oxidation phenomena. Moreover, these molecules can be involved on important effects on host metabolism and health [32,103].

Recently, current research is focused on food-derived peptides and emerging evidence confirmed their strong role in the prevention of oxidative damages. Nevertheless, milk products and fermented milks are recognized as the most common dietary sources of food antioxidants. Indeed, milk protein-derived BPs gained great interest and began to be regarded as a novel class of antioxidants. In particular, ripened chesses (Swiss cheese varieties, Cheddar, Manchego, and Gouda), fermented milks, and yogurts became the subject for numerous studies, which confirmed these products among the major sources of BPs [80,104–106].

Most of the BPs are encrypted in the caseins sequences [48]. Such small motifs are non-active when present in the original proteins but can be liberated after the hydrolysis of protein during GI digestion, microbial fermentation, or enzymatic hydrolysis [39,107]. Parella et al. [108] suggested the idea that antioxidant peptides might be released after a mild heat treatment of milk during cheese manufacturing. The complexity of the production process of FDFs lies in the cheese ripening and proteolysis, identified as the crucial phases for antioxidants formation. In fact, antioxidant peptides are mostly produced by the action of proteases and peptidases released from both starter and non-starter LAB [109–112], residual rennet enzymes, and indigenous milk enzymes such as plasmin [113,114]. Caseins are characterized by a high content of H, P, and Y, widely known for their free radical scavenger activity [115]. Their presence in peptides is a determinant factor in the antioxidant effect [116]. In fact, free radicals are deactivated by peptides containing hydrophobic and aromatic amino acids (such as Y, H, P, W) and selected free amino acids (Y and C) [101,117]. In particular, the presence of Y and W residues in the peptides sequences is considered essential for a significant increase of the antioxidant effect, because of its strong ability to donate a proton [65,118,119]. Actually, their occurrence into the sequences of dipeptides is usually recognized as responsible for antioxidant activities [120]. Histidine displays its antioxidant effects by scavenging free radical species, absorbing active oxygen or through metal ions chelation. Chen et al. [121] described H and P as the most important residues in peptides isolated from a soybean hydrolysate for their inhibitory effect of lipoprotein peroxidation. Kitts and Weiler [122] also reported that the presence of a P or an L residue at the N-terminus position of a HH dipeptide could improve antioxidant activity as well as promote a synergistic effect with other non-peptide antioxidants. Furthermore, M and C, as efficient sulphur hydrogen donors, are

particularly active in quenching free radical species [116,123]. Several studies also confirmed how crucial it is the position of the amino acid residues in the peptide sequence.

The kefir peptides VYPFPGPIPN and QEPVLGPVRGPFPIIV were identified by Eisele et al. [124] as strong antioxidative peptides and weak ACE inhibitors. Other peptide sequences identified in commercial dairy products had previously been described as antioxidant BPs (Table 3). Fragments with the amino acid sequence VKEAMAPK, VLPVPQK, and VPYPQ, obtained from cheddar cheese, milk fermented with *L. rhamnosus,* and yogurt, respectively, exhibited antioxidant activity [125–127]. In addition, the β-casein peptide KVLPVPQ detected in sour milk by Hernandez-Ledesma et al. [119] showed antioxidant activity as occurs with peptide VLPVPQ, previously described by Rival et al. [126]. The octapeptide SKVLPVPQ, isolated and identified in two commercial Spanish fermented milks produced with *L. helveticus* and *S. cerevisiae*, exhibited antioxidant activities [128]. A peptide composed of nine amino acids KIHPFAQTQ isolated from β-casein of buffalo's yogurt has proven to be particularly effective in antioxidant effects [129]. A total of 187 BPs derived from β-casein, αS1-casein, and αS2-casein were detected in cheeses by Pisanu et al. [130]. Among these, nine displayed a strong antioxidant activity, originated from the degradation of β-casein, specifically from the fragment f (194–208), QEPVLGPVRGPFPIL.

**Table 3.** Peptides with antioxidant and antimicrobial activities identified in commercial fermented dairy products.

| Peptide | Fragment | Source | References |
|---|---|---|---|
| ARHPHPHLSFM ● | κ-casein f(96–106) | Yogurt | [55,129] |
| AVPYPQR * | β-casein f(177–183) | Kefir | [70,126] |
| AYFYPE ● | αS1-casein f(143–148) | Yogurt, cheese | [55,64] |
| AYFYPEL ● | αS1-casein f(143–149) | Cheese | [53,54,64] |
| EMPFPK ■ | β-casein f(108–113) | Yogurt | [55] |
| EVFGKEKVN ■ | αS1-casein f(30–38) | Kefir | [70,131] |
| FALPQYLK ● | αs2-casein f(174–181) | Kefir | [70,132] |
| FSDKIAKYIPIQ ■ | κ-casein f(18–29) | Yogurt | [55] |
| GPVRGPFPII ● | β-casein f(199–208) | Fermented milk, yogurt | [119,127] |
| HLPLPL ● | β-casein f(133–138) | Yogurt, fermented milk | [55,105] |
| IPIQY ● | κ-casein f(26–32) | Yogurt | [127] |
| IPIQYVL ● | κ-casein f(26–30) | Fermented milk | [119] |
| KAVPYPQ ● | β-casein f(176–182) | Yogurt | [127] |
| KIHPFAQTQ ● | β-casein f(48–56) | Yogurt | [133,134] |
| KVLPVPQ ● | β-casein f(169–175) | Fermented milk | [119] |
| KVLPVPQK ● | β-casein f(169–176) | Fermented milk | [105,126] |
| LQDKIHP ● | β-casein f(45–51) | Yogurt | [134] |
| PYVRYL * | αs2-casein f(203–208) | Kefir | [132] |
| QEPVLGPVRGPFPII ● | β-casein f(194–208) | Yogurt | [127] |
| QQPVLGPVRGPFPIIV ● | β-casein f(194–209) | Yogurt | [127] |
| RDMPIQ ● | β-casein f(183–188) | Fermented milk | [105,134] |
| RPKHPIK ■ | αS1-casein f(1–7) | Cheese | [58] |
| RPKHPIKHQGLPQEVLNENLLRF ■ | αS1-casein f(1–23) | Kefir | [70] |
| RY ● | Various fragments in αS1-casein, αS2-casein, κ-casein | Fermented milk, cheese | [53,119] |
| RYLG ● | αS1-casein f(90–93) | Cheese | [53,54,64] |
| RYLGY ● | αS1-casein f(90–94) | Cheese | [53,54,64] |
| SDIPNPIGSENSE ■ | αS1-casein f(180–192) | Kefir | [70] |
| SKVLPVPQ ● | β-casein f(168–175) | Fermented milk drinks | [128] |
| STVATL ■ | κ-casein f(141–146) | Yogurt | [55] |
| TVQVTSTAV ■ | κ-casein f(161–169) | Yogurt | [55] |
| VKEAMAPK ● | β-casein f(98–105) | Fermented milk, cheese | [105,125,126] |
| VLNENLLR ■ | αS1-casein f(15–22) | Kefir | [70] |
| VLPVPQK * | β-casein f(170–176) | Fermented milk | [105,126] |
| VPYPQ ● | β-casein f(178–182) | Yogurt | [127] |
| VYPFPGPIPN ● | β-casein A2 f(59–68) | Kefir | [70,124] |
| YQEPVLGPVRGPFPI ■ | β-casein f(191–205) | Kefir | [70] |
| YQEPVLGPVRGPFPIIV * | β-casein f(191–207) | Kefir | [124] |
| YVL ■ | κ-casein f(30–32) | Yogurt | [55] |

● means antioxidant peptide; ■ means antimicrobial peptide; * means antioxidant and antimicrobial peptide.

In conclusion, FDFs may be a rich source of natural antioxidants, which can be consumed or used as natural ingredients to develop functional foods that may enhance the biological value and preservation of food products as well as the healthy conditions of the consumer [122,135–137]. This ability of peptides to inhibit oxidative phenomena by binding or interacting with radicals and transition metals could be a boon to boost human health, although in vivo animal studies and human trials confirming these effects are necessary.

## 2.3. Anti-microbial Peptides Identified in Commercial Fermented Dairy Products

Another interesting property of milk-derived BPs is related to their antimicrobial effect. These peptides show different amino acid composition and mechanism of action. For example, it has been proposed that kappacin form an amphipathic α-helix structure in the space allowing the formation of an anionic pore, increasing the permeability of the membrane of the bacteria [138]. The antimicrobial αS2-casein-derived peptide VYQHQKAMKPWIQPKTKVIPYVRYL was able to permeabilize the inner and outer membrane of *Escherichia coli* and *Staphylococcus carnosus* by initially binding lipopolysaccharide or lipotechoic acid, respectively [139].

Most of the antimicrobial peptides originated from commercial FDFs have been identified in cheeses, yogurt, and kefir (Table 3). The ability of LAB found in fermented dairy products to generate anti-microbial peptides from milk protein hydrolysis may confer a competitive advantage, thus reducing the growth and survival of food-borne pathogens [140].

Kunda et al. [55] found five different antimicrobial peptides in a functional commercial yogurt. Moreover, Dallas et al. [70] identified six antimicrobial peptides released by LAB in kefir. Among them, LAB population in kefir was able to release the antimicrobial peptide isracidin (RPKHPIKHQGLPQEVLNENLLRF) corresponding to the αS1-casein fragment 1–23. This peptide was also identified in Coalho cheese [141] and was able to protect in vivo mice from *Listeria monocytogenes*, *Streptococcus pyogenes*, and *Staphylococcus aureus* infections [142].

Some of the reported peptides displayed bactericidal activity against both Gram-positive and Gram-negative bacteria. For example, the peptides YVL and PYVRYL, respectively isolated from commercial yogurt and kefir, resulted as in vitro growth inhibitors of the Gram-negative pathogenic bacteria *E. coli, Serratia marcescens,* as well as the Gram-positive pathogenic bacteria *Listeria innocua* and *S. carnosus* [143,144]. The peptide VLNENLLR (also known as caseicin B) inhibited the growth of pathogenic bacteria *E. coli* and *Escherichia sakazakii* at a concentration of 0.22 mmol/L.

The β-casein-derived anti-microbial peptides YQEPVLGPVRGPFPI and YQEPVLGPVRGPFPIIV as well as isracidin have been found intact in the GI tract of mini-pig and calf suggesting a possible in vivo role for these bioactive peptides [145,146]. Indeed, Boutrou et al. [147] found that the β-casein-derived peptides VLPVPQK and EMPFPK were released in the human jejunal effluent during GI digestion of milk proteins.

## 2.4. Anti-diabetic Peptides Identified in Commercial Fermented Dairy Products

As reported by the World Health Organization (WHO), about the 90% of diabetes cases are ascribed to T2D and, globally, about 15 million people have T2D, a perspective that could double by 2025 [148–150].

Actual pharmacologic treatments for T2D focus on increasing insulin availability, improving sensitivity to insulin, delaying the absorption of glucose in the GI tract, or stimulating glucose excretion by urines. Dipeptidyl peptidase IV (DPP-IV) has emerged as a new target for the T2D treatment. DPP-IV is a serine endopeptidase located in the GI tract, kidneys, and endothelial layer of the vascular system. It is involved in the regulation of several physiological processes, such as blood glucose homeostasis [151]. DPP-IV is accountable for the quick degradation and inactivation of incretins such as glucagon-like peptide 1 (GLP−1) and glucose-dependent insulinotropic polypeptide (GIP) [152]. The incretin system plays a crucial role in the release of insulin from pancreatic β-cells, in response to high concentrations of blood glucose [150]. DPP-IV inhibitors extend the GLP−1 availability and

improve glucose tolerance in diabetic patients by enhancing the GLP-1 activities [153,154]. Actually, natural peptides and food protein hydrolysates, which could act as potential inhibitors of DPP-IV attracted great interest of the scientific community. Specific peptide sequences, displaying an N-terminal W and/or P at the second position, demonstrated high effectiveness in inhibiting DPP-IV activity. Animal studies performed with food protein hydrolysates demonstrated that an assayed in vitro DPP-IV inhibitory effect is usually correlated to the antidiabetic effects in vivo. Their outcomes also reported that milk proteins and dairy products are especially rich in DPP-IV inhibitory peptides [155,156]. Uenishi et al. [157] identified 46 peptides derived from β-, αS1-, and αS2-casein in commercial gouda-type cheese (Table 4).

**Table 4.** Peptides with in vitro dipeptidyl-peptidase-IV (DPP-IV) inhibitory activity identified in commercial fermented dairy products [1].

| Peptide | Fragment | Source | IC$_{50}$ μmol/L | References |
|---|---|---|---|---|
| WL | α-lactalbumin f(104–105), α-lactalbumin f(118–119) | Yogurt | 44 | [55] |
| LPQNIPPL | β-casein f(70–77) | Cheese | 46 | [157] |
| LPQ | β-casein f(70–72) | Cheese | 82 | [55] |
| VPITPTL | αs2-casein f(117–123) | Cheese | 110 | [157] |
| VPITPT | αs2-casein f(117–122) | Cheese | 130 | [157] |
| LPQNIPP | β-casein f(70–76) | Cheese | 160 | [157] |
| GPFPILV | β-casein caprine f(201–207) | Kefir | 163 | [69] |
| FPGPIPN | β-casein f(62–68) | Cheese | 260 | [157] |
| YP | Various fragments in β-casein, αS1-casein, κ-casein | Yogurt | 658 | [55] |
| YPFPGPIPN | β-casein f(60–68) | Cheese, kefir | 670 | [48,58,73,157,158] |
| PGPIHNS | β-casein f(63–69) | Cheese | 1000 | [157] |
| IPPLTQTPV | β-casein f(74–82) | Cheese | 1300 | [157] |
| PQNIPPL | β-casein f(71–77) | Cheese | 1500 | [157] |
| VPPFIQPE | β-casein f(84–91) | Cheese | 2500 | [157] |

[1] Peptides are classified according to their inhibitory strength; IC$_{50}$ is defined as the amount of peptides (μmol/L) that cause the 50% inhibition of the enzymatic activity.

About the half of these peptides had an X–P structure at their N-termini. Among these, 10 have been tested for their DPP-IV inhibitory effect: VPITPT, LPQNIPP, PQNIPPL, VPITPTL, FPGPIPN, PGPIHNS, IPPLTQTPV, VPPFIQPE, YPFPGPIPN, and LPQNIPPL. The inhibitory activity, expressed as IC$_{50}$ value, of LPQNIPPL (β-CN f70–77) displayed to be the highest recorded effect. These results suggested that LPQNIPPL was the major DPP-IV-inhibitory peptide in cheese ripened for 12 months and played a significant role in in the in vivo inhibition of DPP-IV. The IC$_{50}$ of VPITPTL (β-CN f84–91), VPITPT (αs2-CN f117–122), LPQNIPP (β-CN f70–76), and FPGPIPN (β-CN f62–68) were 110, 130, 160, and 260 μmol/L, respectively. In contrast, although IPPLTQTPV (β-CN f74–82) and VPPFIQPE (β-CN f84–91) and PQNIPPL (β-CN f71–77) have the X–P structure, their IC$_{50}$ values were higher than 1000 μmol/L. YPFPGPIPN, identified in kefir and gouda-type cheese, displayed DPP-IV inhibitory activity but with high IC$_{50}$ value (670 μmol/L). Interestingly, its presence at the intestinal level after consumption of milk and dairy products suggests a high resistance capacity to the GI conditions and a greater probability to be absorbed and detected at the systemic level, available for further in vivo activities [48,58,59,157]. Other shorter peptide motifs were detected in yogurt by Kunda et al. [54]. LPQ and WL were the peptides with the highest inhibitory effect on DPP-IV, with IC$_{50}$ values of about 82 and 44 μmol/L, respectively.

However, the supporting evidences of an in vivo effect of DPP-IV inhibitory peptides in the T2D treatment are very scarce. The study of dietary peptides-mediated DPP-IV inhibition in humans is still in an embryonic phase. Food protein hydrolysates with DPP-IV inhibitory properties were mainly investigated in vitro. Very few studies analyzed serum DPP-IV activity after food administration.

Scarce results about the DPP-IV activity reduction as a consequence of the consumption of protein-rich food are obtained [159]. Moreover, clear indication about the bioavailability of food-derived peptides is limited [160]. Despite that, it is likely that DPP-IV inhibition may be a crucial mechanism in the reported in vivo antidiabetic effects of intact or hydrolyzed food-proteins.

## 3. FDF as Source of BPs-Producing LAB

### 3.1. FDF Microbiota under the Lens of Metagenomics and Genomics

The abundance and diversity of BPs in FDF result from microbiota inhabiting these ecosystems. The LAB population strongly modulates the content, as well as the spatial diversification and temporal distribution of BPs through a balance between consumption and production. The most common LAB responsible for conversion of milk into FDF includes a heterogeneous arena of Gram-positive non-sporulating species belonging to the genera *Lactococcus, Lactobacillus, Streptococcus, Leuconostoc,* and *Enterococcus* [161]. These species are ubiquitous in several niches and, due to their long safe history in food production, the majority of them received the generally recognized as safe (GRAS) status by U.S. Food and Drug Administration (FDA) (http://www.accessdata.fda.gov/scripts/fdcc/?set=GRASNotices) and the qualified presumption of safety (QPS) approval by EFSA. Thirty-six *Lactobacillus* species have currently gained the qualified presumption of safety (QPS) status from EFSA [162]. The microorganisms involved in initial steps of milk fermentation processes, termed as "dairy starters" or "primary starters" (SLAB), have the pivotal function to convert lactose into lactic acid, leading to a fast lowering of pH and redox potential [163]. In cheese-making, SLAB also contribute to flavor development through lactose and citrate metabolism, lipolysis, and proteolysis with the subsequent catabolism of amino acids [164]. Natural or commercial starters are often limited to a few homofermentative LAB species. The mesophilic SLAB group comprises *Lactococcus lactis* ssp. *lactis* and *Lactococcus lactis* subsp. *cremoris*, but undefined natural starters can also contain strains of *Leuconostoc* spp. for the ability to ferment citrate [165]. The thermophilic SLAB, including *L. helveticus, Streptococcus thermophilus,* and *L. delbrueckii* subsp. *bulgaricus*, are most frequently encountered in natural starters for semi-hard and hard cheese-making. As an exception, kefir grains are complex consortia of yeasts (e.g., *Saccharomyces, Candida, Kluyveromyces, Debaryomyces,* and *Torulaspora*) and bacteria (e.g., *Lactobacillus, Lactococcus, Leuconostoc,* and *Streptococcus*) embedded in a polysaccharide matrix, used as a starter to produce a naturally carbonated, slightly acidic fermented dairy product [166].

Apart from yogurt that contains the protosymbiotic species *St. thermophilus* and *L. delbrueckii* subsp. *bulgaricus*, the microbiota of FDFs are often complex bacterial consortia with different species and strains, which drive the development of regional product qualities, as reviewed by Macori and Cotter [167]. In the case of cheese manufacturing, SLAB usually decrease by autolysis after curd acidification, whereas sub-dominant adventitious LAB, termed non-starter LAB (NSLAB), remain metabolically active and become dominant in the ecosystem during cheese ripening. NSLAB are mainly mesophilic facultatively (*L. casei, Lactobacillus paracasei, Lactobacillus plantarum/paraplantarum, Lactobacillus pentosus, L. rhamnosus,* and *Lactobacillus curvatus*) or obligately (*Lactobacillus brevis Lactobacillus fermentum, Lactobacillus buchneri,* and *Lactobacillus parabuchneri*) heterofementative lactobacilli derived from raw milk or cheese manufacturing environment and are pivotal in determining cheese flavor and texture. In some cases, pediococci, such as *Pediococcus acidilactici* and *Pediococcus pentosaceus*, enterococci, such as *Enterococcus durans, Enterococcus faecalis* and *Enterococcus faecium*, and also leuconostocs can be found in NSLAB populations [140].

Recently, metagenomic approaches reveal how FDF, especially cheese, possess fermentation-associated microbial communities more complex than that expected by culture-dependent approaches [167–170]. These communities undergo substantial fluctuations in function of time, space (core vs. surface), and specific combination of abiotic factors (shaping, mixture, salting, and temperature) [171]. On the other hand, the availability of an increasing number of genomes from dairy and non-dairy isolates supports how considerable differences exist at inter- and intra-species level,

resulting in highly different niche adaptability and functionalities. This growing body of knowledge makes it possible to identify or predict functional traits of FDF-associated LAB. Genome analysis showed that dairy lactobacilli retained more genes involved in sugar transportation, proteolysis, and amino acid transportation compared to gut lactobacilli, a signal of adaptation to milk environment [172].

## 3.2. The Renaissance of Interest Towards Artisanal FDF

The market of functional foods is continuously asking for diversification of the range of available products. To meet this need, there is a growing interest to identify new bio-functional strains. Genomics and metagenomics evidences support how dairy LAB are not only important for modifying technological and organoleptic properties of FDF, but also promising candidates for promoting human health. The great inter and intra-species diversity of microbial consortia in FDF makes these matrices interesting bio-reservoirs of novel probiotic candidates. Evidences have been accumulating that FDFs are dietary source of live organisms [37] and that the allochthonous bacteria from FDF can resist stressful GIT conditions [173]. Despite their transient occurrence in the GI tract, dairy LAB can nonetheless influence the resident microbiome and exert host-specific health benefits [14]. In addition to human GIT, potential probiotic candidates have been recently isolated from different kinds of fermented milks [172–176], cheeses [177–182], and kefir. Even if rigorous guidelines define when a single microbe or microbial consortia can be properly qualified as probiotic [183], these studies provided the basis for the development of new functional foods and defined a cohort of strains which will be in future employed as therapeutic adjuvants for in vivo studies. Recently, Sanders and coworkers [184] proposed the concept of "shared core benefits" to explain how probiotic benefits can result from mechanisms conserved within the same taxon, overcoming the idea of strain-specific probiotic effects. Thus, although the FDF microbes cannot be considered probiotic, many of them are evolutionarily highly related to probiotic organisms, and they often share the same molecular mechanisms responsible for health-promoting properties in probiotics [37].

Following a similar trend, FDFs and ripened cheeses, in particular, became relevant sources for LAB suitable to enrich PHMs in food products thanks to the bacterial fermentative activities [185]. Table 5 summarized the main works, which investigated the capability of LAB isolated from FDF to produce BPs in novel food matrices.

Table 5. Main lactic acid bacteria (LAB) strains isolated from FDFs and exploited in developing novel dairy food enriched in BPs.

| Species | Strain | Food Source | Novel Dairy Product | BPs-mediated Activities | References |
|---|---|---|---|---|---|
| *L. helveticus* | 130B4 | Mongolian camel milk | Fermented skim milk | ACE-i | [186] |
| | s6-HTCH, s10-AVCH, s12-AVCH | Mexican Chiapas cheese | Fermented milk | ACE-i | [187] |
| | CM4 [1], CP790 [1] | Japanese sour milk | Fermented milk | ACE-i | [188] |
| | CPN4 [1] | Japanese sour milk | Yogurt | ACE-i | [60] |
| | H9 [1] | Traditional Tibetan kurut (fermented yak milk) | Different kinds of fermented milk, yogurt | ACE-i | [189] |
| | 141, T105 | Polish dairy products | Fermented milk | Antioxidative, opioid, stimulating, hypotensive, immunomodulating, antibacterial and antithrombotic | [190] |
| | ASCC474, ASCC1188, ASCC1315 | Australia dairy products | Fermented milk | ACE-i, immunomodulatory | [191] |
| | DPC4571 | Swiss Cheese whey | Fermented milk; adjunct in Cheddar cheese | ACE-i | [192] |
| | MTCC5463 | Indian dairy product | Honey- flavored fermented milk | Antihypertensive | [193] |
| | LBB BY 21 | Yogurt | Fermented milk | Antimicrobial | [194] |
| | ND01 | Chinese naturally fermented milk | Gouda cheese | ACE-i | [195] |
| | PR4 | Italian cheese | Fermented caseinate | ACE-i | [196] |
| | M92 | Spontaneously acidified milk | Yogurt | ACE-i | [197] |
| *L. casei* | IMAU20411 | Chinese Dairy food | Fermented milk | ACE-i | [158] |
| | Zhang | Koumiss | Cheddar cheese | ACE-i | [198] |
| | LLG | Cheese | Fermented whey drink | ACE-i | [199] |
| | PRA205 | Parmigiano Reggiano cheese | Fermented skim milk, yogurt | ACE-i | [80] |
| | ATCC393 | Dairy food | Fermented milk | ACE-i, antioxidant and anticancer | [200] |
| | ATCC393 | Dairy food | Yogurt with St. thermophilus and L. bulgaricus | | [201] |
| *L. paracasei* | in co-culture with Candida | Comté cheese | Whey drink | ACE-i, reduction in α-lactalbumin and β-lactoglobulin | [202] |
| | MTCC 5945 (NS4) | Fermented milk | Fermented camel milk | ACE-i | [203] |
| | PTCC 1637 | Camel sour milk | Fermented camel milk | ACE-i and antioxidant | [204] |
| *L. plantarum* | 55 | Portuguese raw ewe milk semisoft cheeses | Fermented cow skim milk | anti-inflammatory, antihemolytic and antioxidant | [205] |
| | KX881772, KX881779 | Camel milk | Fermented cow camel milk | ACE-i, antiproliferation | [206] |
| | KX881772, KX881780 | Camel milk | Low-fat akawi cheese | ACE-i, antioxidant | [207] |
| | 69 | Mongolian fermented cow milk | Fermented goat milk | ACE-i | [208] |
| | LMG18024 | Buffalo milk | Fermented skim milk | ACE-i | [209] |
| | BGPV2–45a, BGBUK 2–5, BGGA-8 | Homemade cheese | Fermented skim milk | ACE-i | [209] |

**Table 5.** *Cont.*

| Species | Strain | Food Source | Novel Dairy Product | BPs-mediated Activities | References |
|---|---|---|---|---|---|
| *L. delbrueckii* subsp. *bulgaricus* | SS1 | Dairy product | Fermented milk | ACE-i | [210] |
| | ACA-DC 87 | Greek sheep milk yogurt | Greek yogurt and feta cheese | ACE-i | [211] |
| | b38 | Yogurt | Fermented milk | Antimicrobial | [194] |
| | CRL 581 | Argentinian hard cheese | Fermented milk | ACE-i, anti-inflammatory | [212] |
| *Lc. lactis* subsp. *lactis* | NRRL B-50571, NRRL B-50572 | Cheese | Fermented cow milk | ACE-i | [213] |
| | Q1, Q5 | Chihuahua cheese | Fermented milk | ACE-i | [214] |
| | ESI197, M21, P21 | Manchego cheese | Manchego cheese | ACE-i | [215] |
| | KX881782 | Camel milk | Fermented camel milk | ACE-i, antiproliferation | [206] |
| *Lc. lactis* subsp. *cremoris* | FT4 | Dairy product | Fermented milk | ACE-i | [210] |
| *Leu. lactis* | PTCC1899 | Traditional fermented camel milk (Chal) | Fermented milk | ACE-i, antioxidant | [176,216] |
| *St. thermophilus* | ACA-DC 835 | Greek Cow milk yogurt | Greek yogurt and feta cheese | ACE-i | [211] |
| | LMD-9, PB302, PB385, CNRZ404, ATE19PB8, Y4, HAD8a, CNRZ1066 | Yogurt | Fermented cow milk | ACE-i, immunomodulating and antimicrobial | [217] |
| | CNRZ445 | Cheese | Fermented cow milk | ACE-i | [217] |
| | ATCC19258 | Pasteurized milk | Fermented cow milk | ACE-i | [217] |
| | in co-culture with L. delbrueckii subsp. bulgaricus | Yogurt | milk–juice beverage with fermented sheep milk and strawberries | ACE-i | [218] |
| *L. acidophilus* | L10 | Dairy food | Fermented milk | ACE-i | [219] |
| | NCDC-15 in co-culture with St. thermophilus NCDC167 | Lassi | Fermented buffalo milk | ACE-i, immunomodulatory, antioxidant, opioid and cytomodulatory | [220] |
| *E. faecalis* [2] | CECT 5727, CECT 5728, CECT 5826, CECT 5827 | Raw milk | Fermented skim milk | ACE-i | [221] |
| | 43 | Brazilian artisanal ripened cheese | Fermented cow milk | nr | [222] |

**Table 5.** *Cont.*

| Species | Strain | Food Source | Novel Dairy Product | BPs-mediated Activities | References |
|---|---|---|---|---|---|
| *Kefir grains* | Lactobacilli and yeast consortium | - | Fermented cow milk | antihypertensive, antimicrobial, immunomodulatory, opioid and anti-oxidative | [70] |
| | Lactobacilli and yeast consortium | - | Kefir | ACE-i, antimicrobial | [223] |
| *L. reuteri* | KX881777 | Camel milk | Fermented cow camel milk | ACE-i, antiproliferation | [206] |
| *P. acidilactici* | 90 | Brazilian artisanal ripened cheese | Fermented cow milk | nr | [222] |
| Undefined LAB species | s6-HTCH, s10-AVCH, s12-AVCH | Chiapas cream cheese | Fermented milk | ACE-i, antioxidant | [187] |
| *W. viridescens* | 111 | Brazilian artisanal ripened cheese | Fermented cow milk | nr | [222] |

[1] Calpis Food Industry Co. Ltd. (Tokyo, Japan) selected cultures; [2] *Ent. faecalis* is not recommended for the QPS list (EFSA 2016).

The majority of these works employed a so-called "reverse food engineering" strategy, which entailed the following three steps: (i) The isolation of LAB from traditional FDFs; (ii) the screening of LAB for the ability to release BPs; and (iii) the exploitation of the best BPs producer strains for innovation in dairy food fermentation (Figure 2).

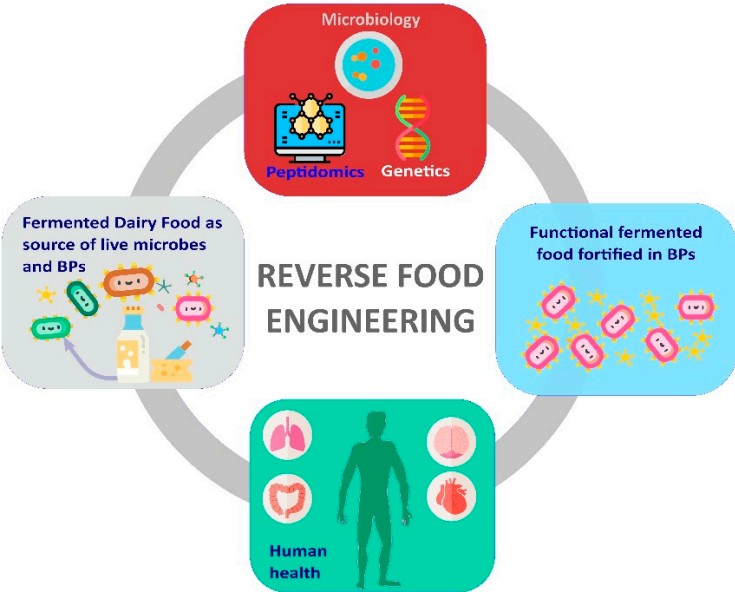

**Figure 2.** Overview of reverse food engineering approach. Reverse food engineering exploits fermented dairy foods (FDFs) as bio-reservoirs of microbes producing BPs and pro-health molecules and performs multidisciplinary analysis to improve content of BPS and healthy microbes in hyperfood.

*L. helveticus* exploited in hypotensive drinks such as Evolus (Valio Ltd., Valio, Finland) or sour milk like Calpis (Calpis Food Industry Co. Ltd., Tokyo, Japan) were isolated for spontaneous fermented milk or whey [188,224]. *L. helveticus* strains isolated from Mongolian fermented milks showed different BPs releasing abilities than *L. helveticus* strains from Western countries [225]. Brazilian cheeses, such as Manchego and Coalho cheeses [215], were found to be source of highly proteolytic LAB suitable to produce drinkable milk fortified in BPs. Similarly, Rutella et al. [80] successfully used BPs-producer NSLAB *L. casei* and *L. rhamnosus* from Parmigiano Reggiano ripened cheese to develop a yogurt enriched in ACE-i VPP and IPP.

## 4. LAB Proteolytic System

LAB have a complicated proteolytic apparatus, which is able to hydrolyze milk proteins resulting in the release of oligopeptides and then free amino acids. The most-studied proteolytic system was from *Lc. lactis* and includes three main constituents: a cell-envelope proteinase (CEP), transport systems for short peptides and amino acids, and a multitude of intracellular peptidases. Milk proteins are initially hydrolyzed to oligopeptides by CEP. Afterwards, the released peptides are transported inside the cells by specific transporters including oligopeptide transporters (Opp) and di-tripeptide transporters (Dpp, DtpP, and DtpT). Finally, peptidases convert these peptides into free amino acids, which are essential for the growth of LAB [215] (Figure 3A).

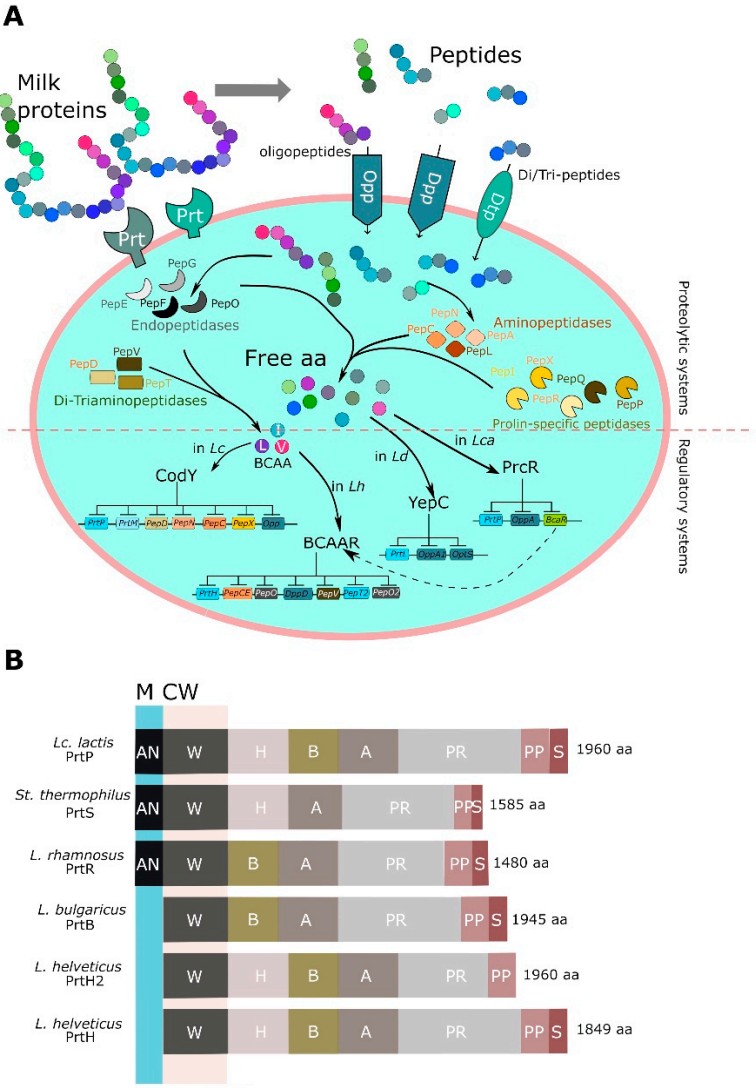

**Figure 3.** Comparative analysis of proteolytic systems in *Lactococcus lactis* and *Lactobacillus* spp. (**A**) Main effectors of proteolytic machinery and regulatory systems. (**B**) Multi-domain structure of prt proteinases isolated from *Lactococcus lactis*, *Streptococcus thermophilus*, and *Lactobacillus* spp. Abbreviations: BCAA, branched chain amino acids; Lc, *Lactococcus lactis*; Lb, *Lactobacillus bulgaricus*; M, membrane; CW, cellular wall; S, signal; PP, pro-domain; PR, proteolytic domain; A, A domain; B, B domain; H, helix domain; W, cell wall domain; AN, anchoring domain. I domain was omitted for brevity.

Several evidences support that the lactococci and lactobacilli proteolytic systems are comparable in their components and modes of action in terms of converting milk proteins (primarily caseins) into oligopeptides and free amino acids. The most proteolytic species *L. helveticus* is one of the most nutritionally fastidious LAB, needing 14 exogenous amino acids [226]. This auxotrophy is counterbalanced by a potent proteolytic system, which fulfils the amino acid requirements of the *L. helveticus* cell [227]. Comparative genome analyses support that proteolytic enzymes are highly variable at inter and intra-species level, mainly for CEP-encoding genes, determining a huge variability in BPs release among different LAB strains. Therefore, the proteolytic activity of a given strain and the resulting BP profiles in food matrix finally depend on its CEP and peptidase patterns, on gene variations, and on the environmental conditions, which modulate proteolytic gene expression [228].

### 4.1. Proteases

CEPs are serine proteases belonging to the family of subtilisine, which catalyze the breakdown of caseins into oligopeptides. Based on the substrate specificity, CEPs were classified in three types: PI-type CEPs primarily hydrolyze β-casein; PIII-type CEPs are able to degrade all caseins, including αs1-, κ-, and β-casein; PI/PII-type CEPs hydrolyze both β-casein and, to a lesser extent, αs1-casein [229]. Generally, LAB synthesize CEPs as preproteins of ~2000 residues and, in their final arrangement, they include seven functional domains (from N to C-terminus): S, PP, PR, A, B, H, and W. Among these, PP is the pre-pro domain, which includes a typical signal sequence required for secretion, while PR identified the catalytic protease domain with a small inserted (I) domain, which can modify the prt-specificity. The A domain has unknown function, while the B domain probably plays a stabilizing role; the H domain is a helical spacer, whereas the W domain is a hydrophilic cell wall spacer or attachment domain [230]. Three distinct anchoring mechanisms were noted: A SLAP domain (S-layer anchoring domain), particularly present in the *L. delbrueckii* sub-clade, is responsible for the non-covalent interactions; a LPXTG motif involved in the covalent linkage to peptidoglycan; and a LPXTG motif-derivative. In some cases, like in *L. helveticus,* the typical CEP anchoring domain was not found at C-terminus and the sequences ended just before the typical start of the anchoring domain motif. Based on the anchoring system and the possibilities of various domain combinations, different CEPs were characterized in different LAB species, including PrtP from *Lc. lactis*, *L. casei* and *L. paracasei*, PrtH from *L. helveticus*, PrtR from *L. rhamnosus*, PrtS from *S. thermophilus*, and PrtB by *L. delbrueckii* subsp. *bulgaricus*, as reviewed by Liu et al. [231]. The main features of most relevant CEPs in LAB species are depicted in Figure 3B.

More recently, comparison of 213 genomes from *Lactobacillus* species revealed a more extensive diversity in CEP characteristics, with a total of 60 CEPs identified [232]. Most LAB possess only one type of CEP, while *L. helveticus* strains have at least two different prt paralogs, namely PrtH and PrtH2 [233]. In a survey of *prt* genes in different *L. helveticus* strains, Genay et al. [234] revealed that PrtH2 is almost always present. In some *L. helveticus* strains, like CNRZ32, prtH3, and prtH4 paralogs were also present [235]. PrtH1, PrtH2, PrtH3, and PrtH4 were expressed in a strain-dependent manner by *L. helveticus* [226]. In addition, *L. helveticus*, as other LAB species, requires a maturation proteinase named PrtM for the activation of CEP. In *L. helveticus* strain CNRZ32 two PrtMs (PrtM1 and PrtM2) have been identified [236]. Further studies [234,235,237] stated that PrtM is required for the activation of PrtH and PrtM2 and plays an important function in the activation of further *L. helveticus* CEP paralogs.

### 4.2. Transport System and Peptidases

Peptides liberated by the hydrolytic action of CEPs are then imported by several transport systems within the cells, where they can be further cleaved by intracellular peptidases, releasing amino acids. Different transport apparatuses are implicated in the internalization of peptides in LAB: The oligopeptide permease system (Opp), the dipeptide permease system (Dpp), and transporter of di/tripeptides (DtpT).

The Opp system falls into the superfamily of ATP-binding cassette (ABC) and is able to transport oligopeptides from 4 up to 35 amino acids [238–240]. Biochemical and genetic characterization of the Opp system in *Lc. lactis* revealed that it is composed by five different proteins: The protein OppA that is responsible for the oligopeptides binding, the integral membrane OppB and OppC proteins, and the nucleotide-binding OppD and OppF proteins [238,240]. The genes encoding for the Opp system domains are located upstream the *pepO* gene, encoding for the endopeptidase PepO [237]. The system was also genetically characterized in *L. delbrueckii* subsp. *bulgaricus*, where an extra homolog gene *oppA2* is present [241]. The Opp system of other LAB was not widely investigated, but Liu et al. [231] found the genes for the Opp system in several strains of *L. acidophilus*, *L. brevis*, *L. casei*, *L. rhamnosus*, *L. johnsonii*, *L. gasseri*, and *L. helveticus*. Most of the sequenced strains possessed more copies of the oligopeptides-binding protein OppA. Strains belonging to the *L. acidophilus* group, such as *L. acidophilus*,

*L. johnsonii,* and *L. gasseri,* possessed the highest number of OppA genes [231]. However, at present, it is not clear whether all these paralogs are functional.

The DtpT transport system belongs to the family of the PTR (proton motive force-driven peptide transport system) [242]. DtpT, firstly identified in *Lc. lactis*, is a transmembrane protein of 12 residues with the C- and N- terminal facing the cytoplasm and *DtpT* gene positioned after the *pepN* gene, which encode the aminopeptidase PepN. This system shows a higher affinity for hydrophilic and charged di- and tripeptides [242]. Other LAB, such as *L. helveticus,* possess a DtpT system with similar specificity as that found in *L. lactis* [243]. Finally, Liu et al. [231] found the *DtpT* gene in several LAB strains including some species of interest for dairy fermentation such as *L. acidophilus*, *L. brevis*, *L. casei*, and *L. rhamnosus*.

Focaud et al. [244] firstly identified in *Lc. lactis* a third transport system (Dpp) belonging to the ABC superfamily and with a preference for di- and tripeptides that contained hydrophobic branched-chain amino acids. The Dpp system is organized as the Opp system with the subunit DppA, which is responsible, in analogy with OppA, for the binding of di- and tripeptides [245]. In both the systems, the peptide-binding proteins (DppA and OppA) are fixed to the membrane thorough a modified cysteine residue at the N-terminal and deliver peptides to their cognate membrane complexes [246]. The Dpp system appeared to be widely distributed among the genomes of *Lactobacillus* species with the *DppA* gene being often present in multiple copies [231].

Following the internalization, numerous cytoplasmic peptidases cooperate to degrade the casein-derived peptides with different and partly overlapping specificities [236,247]. The LAB intracellular peptidase can be divided in four groups depending on their specificity: Aminopeptidases (PepN, PepC, PepA, PepM, PepP, and PepX), endopeptidases (PepE, PepG, PepF, and PepO), and di/tri-peptidases (PepI, PepL, PepQ, PepR, PepD, PepV, and PepT) (Figure 3A).

Endopeptidases are probably the first enzymes involved in the processing of peptides transported inside the cells. PepG, PepE, and its paralog PepE2 are cysteine-peptidases, whereas PepF and PepO have been classified as metallopeptidases [248–252]. At least in *L. helveticus*, two paralogs of PepO, named PepO2 and PepO3, with different specificity have been characterized [250,253]. LAB endopeptidases are able to hydrolyze, with different specificity, oligopeptides from 5 to 35 amino acid residues but they were not able to act on intact caseins [236,247].

Other peptidases able to act on oligopeptides are the aminopeptidases PepN, PepC, PepA, PepM, and PepP that were characterized in diverse dairy LAB species, such as *L. casei*, *L. helveticus*, *L. bulgaricus,* and *Lc. lactis* [236,247,254–257]. The metallopeptidases PepA and PepM have very defined substrate specificities and are able to remove the N-terminal amino acids N/E and M, respectively. On the contrary, the metallopeptidase PepN and the cysteine-peptidase PepC have very broad substrate specificity [236,247]. The metallopeptidase PepP is active on oligopeptides from three to nine amino acids length and is characterized for their ability to hydrolyze the N-terminal amino acid when the residue in second position is P [258,259]. An additional aminopeptidase active on oligopeptides of three to seven amino acids is PepX. PepX activity has been detected in several LAB strains isolated from FDFs such as *L. rhamnosus*, *L. casei*, *L. helveticus*, *L. delbrueckii* subsp. *bulgaricus,* and *Lc. lactis* [236,247,260,261]. PepX is a prolyl dipeptidyl aminopeptidase able to liberate X-P di peptides from the N-terminus of an oligopeptides [236,247].

Di- and tripeptides generated by the action of endopeptidases and aminopeptidases are further cleaved in the individual amino acids by the action of di/tri-peptidases. The tri-peptidase PepT and the di-peptidases PepD and PepV are characterized for a broad range of substrate specificity with preference for peptides containing hydrophobic amino acids [236,247,262,263]. They have been characterized in several LAB strains of paramount importance in dairy fermentation such as *L. casei*, *L. helveticus*, *L. delbrueckii* subsp. *bulgaricus, L. sake,* and *Lc. lactis* [236,247,262,263]. With the exception of PepL, which releases the amino acid L at the N-terminus from di- and tripeptides, the others characterized di/tri-peptidases (PepI, PepQ, and PepR) are classified as proline peptidases, each with its own specificities [264–266]. PepI and PepR are proline aminopeptidase, which release the amino acid

P from di- and tripeptides and dipeptides, respectively. Instead, PepQ is a proline carboxypeptidase, which releases the residue P from the C-terminus of a dipeptide. These peptidases are widely distributed among LAB strains and have been characterized in *L. rhamnosus*, *L. casei*, *L. helveticus*, *L. delbrueckii* subsp. *bulgaricus*, and *Lc. lactis*.

Comparative genomics studies support that the cytoplasmic peptidases PepN, PepC, and PepX were commonly found among *Lactobacillus*, encompassing species of importance in dairy fermentation [231,267]. Genes encoding for some peptidases, such as PepC, PepE, PepD, and PepO, were detected as multiple copies in LAB dairy strains, suggesting that these genes could be involved in adaption to habitats rich in proteins and peptides [231,267].

The heterogeneity in the presence of peptidases is also observed at intra-species level, where co-specific strains diverge for the type and number of transport system and peptidases. Comparative analysis of four *L. delbrueckii* subsp. *bulgaricus* genomes (ATCC 11842, BAA-365, 2038, and ND02) showed that the highest number genes encoding proteases, peptidases, and different transport systems were found in strain ND02. With regard to the strain 2038, two cell surface peptidases PepD4 and En1A were found as whole genes, suggesting that this strain is characterized by a stronger proteolytic ability, potentially producing higher amounts of free amino acids than the other strains [231]. All four sequenced strains are equipped with two complete Opp transport apparatuses, but the number and arrangement of the substrate binding protein OppA was different. The industrial strain 2038 was characterized by the highest number of genes encoding the OppA protein [231].

Intracellular peptidases have an important role in cheese maturation. For example, PepX and PepN are partially accountable for the release of volatile aroma-active molecules from milk proteins during cheese ripening [268]. Some other peptidases, such as PepE, PepN, and PepO (and its paralogs PepO2 and PepO3), have been found to hydrolyze the peptide corresponding to the β-casein fragment 193–209, which confers the major bitter taste defect in Gouda and Cheddar cheeses [250,269,270].

Recently, the contribution of *L. helveticus* intracellular peptidases to the release of the antihypertensive lactotripeptides IPP and VPP, from the peptides released by the action of PrtP on β-casein, has been demonstrated. A key enzyme for the processing of VPP and IPP, with high homology to the endopeptidase PepO, has been identified and characterized in *L. helveticus* CM4 strain [271]. This endopeptidase can catalyze the hydrolysis of the C-terminal amino acid residues form the peptides VPPFL and IPPLT, releasing VPP and IPP, respectively. Some other aminopeptidases such as PepC and PepX may be involved in the N-terminal processing of VPP- and IPP-containing peptides [272,273]. However, to obtain more detailed information on the processing involved in VPP and IPP production, it should be of paramount importance to study the release of the lactotripeptides or precursors in *Lactobacillus* strains expressing or disrupting each specific peptidase gene.

## 4.3. Regulation of Proteolytic Systems

Proteolytic genes, especially *prt* ones, are generally repressed in cells growing in nitrogen sources-enriched medium. Regulatory mechanisms that modulate the proteolytic machinery have been well studied in *Lc. lactis*, where the GTP-binding protein CodY is implicated in the repression of various genes (prtP/prtM, opp-pepO1, pepD, pepN, pepC, and pepX) of the proteolytic system in consequence of the availability of branched-chain amino acids (BCAA) [274]. In *Lc. lactis* CodY also regulates aminotransferases AraT and BcaT, which play a pivotal role in regulating the intracellular pool of amino acids [275].

In lactobacilli, no CodY-like proteins were found until now, such that the knowledge about the regulation of the genes encoding proteinases and peptidases remained scarce. Only recently, three works partially elucidated the regulatory mechanisms of proteolysis in *L. helveticus*, *L. casei*, and *L. delbrueckii* subsp. *lactis*. In *L. helveticus* CM4, transcriptional negative regulation of the proteolytic apparatus in response to amino acids is mediated by a BCAA-responsive transcriptional regulator (BCARR), which reduces the production of antihypertensive peptides by CM4 when amino acids are available in the surrounding medium [276]. The DNA-binding proteins YebC and PrcR are responsible

for extracellular peptide-dependent repression of the genes of the proteolytic system in *L. delbrueckii* subsp. *lactis* CRL 581 and *L. casei* BL23, respectively [228,277] (Figure 3B). Differently from *L. helveticus* and *Lc. lactis*, both these regulators are unresponsive to BCAA. Interestingly, PrcR was found to negatively regulate the BcaR gene encoding for a protein orthologous to BCARR, suggesting that PrcR could be the master regulator of a regulatory network responsive to amino acid starvation more complex than that found in in *L. helveticus* [228]

## 5. Conclusions

Collective data from FDF peptidomics and LAB genomics and metagenomics represent important steps to realize the knowledge-based development of food enriched in BPs. The insights into the patterns of BPs in FDF and the microbiota responsible for their release enable the exploitation of FDF as resource for isolating novel potential BP producer strains. At the same time, knowledge on genes involved in BP release phenotype can drive and shorten the screening of the most interesting proteolytic candidates, opening the possibilities for development of products with diverse or enhanced BP patterns and broadens the overall portfolio offered to the final customer. This approach, here referred to as reverse food engineering (Figure 2), will allow the improvement of knowledge on the strict but still poorly explored link between diet and health.

**Supplementary Materials:** Supplementary materials can be found at http://www.mdpi.com/2311-5637/5/4/96/s1.

**Author Contributions:** Conceptualization, L.S.; methodology, D.T., S.M. and L.S.; writing—review and editing, D.T., S.M. and L.S.

**Funding:** This research was funded by the 2017 grant provided by Consorzio del Formaggio Parmigiano Reggiano.

**Conflicts of Interest:** The authors declare no conflicts of interest. The funders had no role in the design of the study; in the collection, analyses, or interpretation of data; in the writing of the manuscript, or in the decision to publish the review.

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
