# Peer review of "Bioprospecting for Bioactive Peptide Production by Lactic Acid Bacteria Isolated from Fermented Dairy Food"

_fermentation, doi:10.3390/fermentation5040096_

Round 1
Reviewer 1 Report
The article is very long. Authors should shorten less important fragments, e.g. explanation of disease mechanisms (line 133-143), etc. to help readers understanding the content of the article.
Detailed comments:
Line 44: The authors should underline, that the definition of the functional food they took into consideration is different from usually recognized.
Line 52: The source of the “hyperfood” definition is needed.
Line 54: What is the source of figure 1?
Line 406: In my opinion, the scheme is unnecessary, it does not bring relevant information. Abbreviations should be explained in the text.
Line 890: This citation contains forenames and shortcuts of the surname. Should be: Cichosz, G., Czeczot, H., etc.
Author Response
Detailed response to criticisms arisen by Reviewer 1
Reviewer 1 (R1)
1) The article is very long. Authors should shorten less important fragments, e.g. explanation of disease mechanisms (line 133-143), etc. to help readers understanding the content of the article.
Answer. We thank the R1 for this suggestion. We shortened parts that detailed the disease mechanisms eg. omitted lines 140-144. Furthermore, we deleted other parts thought the main text.
Detailed comments:
2) Line 44: The authors should underline, that the definition of the functional food they took into consideration is different from usually recognized.
Answer: We modified the “functional food” definition to better address the original Japanese statement approved in 1984 in the legal frameshift “Foods for Specific Health Uses (FOSHU)”.
3) Line 52: The source of the “hyperfood” definition is needed.
Answer: We added the reference about hyperfood. The numbering of References shifted according to this modification.
Veselkov, K.; Gonzalez, G.; Aljifri, S.; Galea, D.; Mirnezami, R.; Youssef, J.; Bronstein, M.; Laponogov I. HyperFoods: Machine intelligent mapping of cancer-beating molecules in foods. Sci. Rep. 2019, 9, 9237.
4) Line 54: What is the source of figure 1?
Answer: We did ourselves the Figure 1.
5) Line 406: In my opinion, the scheme is unnecessary, it does not bring relevant information. Abbreviations should be explained in the text.
Answer: We deleted Figure 2.
Line 890: This citation contains forenames and shortcuts of the surname. Should be: Cichosz, G., Czeczot, H., etc.
Answer: We corrected the error.
Reviewer 2 Report
The manuscript encompasses an intensive review of the topic in the field, with substantial support from references. It would greatly enhance the takeaway message for readers if the authors can add a summary of how solid the scientific evidence is so far for each of the functions. i.e. anti-hypertensive, anti-oxidant, anti-microbial and anti-diabetic. ( in table or so).
Here are a few minor edits suggested:
L131-139: bring the caption and table together for Table 2
L208: remove the first "T"
L293: 2.2=>2.3 ?
L322: add 2.4 ?
L604: a dipeptide => and dipeptide?
Author Response
Detailed response to criticisms arisen by Reviewer 2
Reviewer 2 (R2)
1) The manuscript encompasses an intensive review of the topic in the field, with substantial support from references. It would greatly enhance the takeaway message for readers if the authors can add a summary of how solid the scientific evidence is so far for each of the functions. i.e. anti-hypertensive, anti-oxidant, anti-microbial and anti-diabetic. (in table or so).
Answer: We know that the high number of evidences on biofunctionalities of BPs may reduce the understanding. However, we would avoid to add further Table in order to address the criticism of Reviewer 1 to reduce the length of manuscript. Please consider that the main text includes anti-hypertensive peptides with in vivo evidences of bifunctionalities.
Here are a few minor edits suggested:
2) L131-139: bring the caption and table together for Table 2
Answer: Caption and Table 2 were together in our docx manuscript.
3) L208: remove the first "T"
Answer: Done
4) L293: 2.2=>2.3 ?
Answer: Done
5) L322: add 2.4 ?
Answer: Done
6) L604: a dipeptide => and dipeptide?
Answer: Done